# Pre-Slaughter Stunning of Farmed Atlantic Halibut in CO_2_-Saturated Seawater: Assessment of Unconsciousness by Electroencephalography (EEG)

**DOI:** 10.3390/ani13121993

**Published:** 2023-06-15

**Authors:** Daniel Santiago Rucinque, Hans van de Vis, Henny Reimert, Bjørn Roth, Atle Foss, Cesar Augusto Taconeli, Marien Gerritzen

**Affiliations:** 1Faculdade de Zootecnia e Engenharia de Alimentos, Laboratório de Aquicultura, Universidade de São Paulo, Av. Duque de Caxias Norte 222, São Paulo 13635-900, Brazil; 2Wageningen Livestock Research, P.O. Box 338, 6700 AH Wageningen, The Netherlands; 3Department of Processing Technology, Nofima, 4068 Stavanger, Norway; 4Akvaplan-niva, Fram Centre, 9296 Tromsø, Norway; 5Department of Statistics, Federal University of Paraná, Curitiba 81531-980, Brazil; cetaconeli@gmail.com

**Keywords:** fish welfare, humane slaughter, unconsciousness, EEG

## Abstract

**Simple Summary:**

The World Organization for Animal Health recommends the use of stunning methods before the slaughter of farmed fish destined for human consumption. The use of carbon dioxide (CO_2_) is not recommended by the World Organisation for Animal Health for the stunning of farmed fish at pre-slaughter. However, its use continues to be common in the world. CO_2_ stimulates chemoreceptors, causing aversion and escape attempts when fish are immersed in CO_2_-saturated water. The use of CO_2_ in seawater with oxygenated water is allowed for pre-slaughter of halibut in Norway. Hence, there are no studies assessing loss of consciousness in halibut exposed to this stunning method. The best approach to determine unconsciousness is by measuring brain electrical activity through EEG.

**Abstract:**

As fish welfare becomes a growing concern, it is important to ensure humane treatment during slaughter. This study aimed to assess the onset of unconsciousness in Atlantic halibut immersed in CO_2_-saturated seawater through electroencephalography (EEG). Of the 29 fish studied, 10 exhibited escape attempts, indicating aversion to CO_2_-saturated water despite its oxygenation. EEG signals showed four distinct phases: transitional, excitation (high amplitude–high frequency), suppressed, and iso-electric phases. The onset of the suppressed phase, indicative of unconsciousness, occurred on average 258.8 ± 46.2 s after immersion. The spectral analysis of the EEG signals showed a progressive decrease in median frequency, spectral edge frequency, and high frequency contribution, which corresponded to the gradual loss of consciousness. The study concludes that CO_2_-saturated water is not recommended for pre-slaughter handling of halibut due to the extended time required for the onset of unconsciousness and the observed aversive behaviour. Ensuring humane treatment during slaughter is important for addressing public concern and safeguarding fish welfare in all stages of production.

## 1. Introduction

Global production from aquaculture of fish, crustaceans and molluscs continues to grow and reached 88 million tonnes in 2020. The capture production was 90.3 million tonnes, a decrease of 2.06 percent compared with 2019. The global finfish aquaculture production was 57.5 million tonnes in 2020 [1]. According to the Federation of European Aquaculture Producers the European production of finfish farming production was 2.5 million tonnes in 2020. In 2020, 1500 tonnes of farmed Atlantic halibut (*Hippoglossus hippoglossus*) were produced in Norway [2]. Atlantic halibut is a species of high value in the market that is likely to play a more important role in European aquaculture production [3].

Concern about fish welfare is increasing in Western countries. A study of 2.147 Norwegian citizens showed that they are concerned about fish welfare [4]. Even though levels of public knowledge and concern about fish welfare issues are lower when compared to terrestrial species, society calls for improvement of animal farming practices. For example, Norway has specific regulations for the humane slaughter of farmed fish destined for human consumption [5,6]. However, the use of carbon dioxide (CO_2_) in seawater is allowed for the stunning of halibut, but not for Atlantic salmon. Additionally, the World Organisation for Animal Health—OIE recommends practices to safeguard farmed fish welfare at slaughter [7]. To protect animals at slaughter, they should be rendered unconscious and insensible by stunning to avoid pain, fear or distress prior to slaughter, which is a general provision in the EU legislation to protect animals at slaughter [8]. Hence, to protect fish at slaughter, two steps should be applied. The first is stunning, which leads to loss of consciousness (LOC) and sensibility. The second step is killing by bleeding out or the application of another method to induce death while the fish is still unconscious. It should be noted that both steps may be achieved by the application of only one method. Nevertheless, Annex 1 of the European relation regarding the protection of animals at the time of killing does not mention any specifical requirement to stun/kill fish, which differs from the specific requirements to use head-only electrical stunning in cattle, ovine, caprine, chickens and turkeys, as well as for electrical waterbath stunning of poultry.

Methods to avoid or minimize pain, fear and distress at slaughter of fish comprise electrical stunning, percussion or the use of permitted anaesthetics in water [9]. The European Food Safety Authority [10] recommends assessing stunning and killing methods through electroencephalography (EEG) [11]. EEG registrations in fish are supplemented by behavioural observations [12]. The electrical activity recorded on EEG can be classified into Delta (<4 Hz), Theta (4 to 8 Hz), Alpha (8 to 13 Hz), Beta (13 to 32 Hz) and Gamma (32 to 45 Hz) frequency bands. When the main power contribution of the EEG is in the Alpha, Beta and Gamma frequency bands (8–45 Hz), the animal is considered to be conscious, whereas when the power contribution shifts to lower frequencies, Theta and Delta, the animal is considered to be unconscious [13]. It should be noted that no firm conclusions on LOC can be drawn based on observation of behaviour only [9]. Fourier transformation boils down to spectral analysis, as EEG signals are dissected in its component spectra. In principle, such a method allows for the detection of underlying sinus waves that in their sum result in the complex wave detected [14]. Further, spectral analysis on EEG traces is useful to identify different states of consciousness. Spectral variables can be calculated from EEG as total power (Ptot), median frequency (F50) and spectral edge frequency (SEF). Ptot is defined as the total area under the power spectrum curve; F50 represents the median frequency of the power spectrum curve and SEF represents the frequency where 95% of the power spectrum curve is located [14]. F50 is more sensitive to changes in lower frequencies, whereas SEF is more sensitive to shifts towards higher frequencies. It is well recognized that increases in EEG Ptot and associated decreases in both F50 and SEF are correlated with clinical signs of LOC and anaesthesia [15,16,17].

One of the first reports of the use of carbon dioxide (CO_2_) in water for the purposes of anaesthesia and transport of fish was in carp (*Cyprinus carpio*) [18]. The authors observed drops in the water pH from 7.5 to 5.0 when the tank was flushed with CO_2_ and active movements were observed at the initial phase of anaesthesia [18]. In other experiments, CO_2_ in anoxic water was used for the stunning of Atlantic salmon (*Salmo salar*) [11,19,20] and rainbow trout (*Oncorhynchus mykiss*) [21]. Atlantic salmon and rainbow trout showed strong aversion for at least 30 s after immersion in carbon dioxide, and even durations of more than three minutes have been recorded [22]. For this reason, CO_2_ stunning without aeration or oxygenation has been banned in Norway. In Norway, halibut is stunned in well-oxygenated water flushed with CO_2_ until a pH of approximately 5 is obtained. In practice, the halibuts are netted or pumped into a tank with CO_2_-saturated water and are left in the tank until all movement stops. They are then removed from the water and exsanguinated [19].

At present, assessments of effectiveness of CO_2_ stunning of halibut in well-oxygenated seawater via registration of EEGs have not been performed. Hence, it is not known whether the method applied results in loss of consciousness with avoidable pain, fear or distress in halibut. Therefore, the aim of this study was to assess the onset of unconsciousness through EEG assessment of Atlantic halibut (*Hippoglossus hippoglossus*) immersed in CO_2_ saturated seawater.

## 2. Materials and Methods

### 2.1. Animals and Immersion in CO_2_

This study was approved on 16 May 2015 by the Norwegian authority (Mattilsynet), licence number 2015/108619.

Thirty farmed halibut with an average weight of 2.33 ± 0.56 kg were provided by a commercial halibut producer in Norway, in autumn. Prior to the experiment, fish were kept in holding tanks containing aerated seawater with a pH of 7.8 and at a temperature of 8.7 °C.

Prior to immersion, the tank containing seawater with a temperature of 8.6 °C was flushed with carbon dioxide until a pH of 5.2 was obtained. Typically, commercial stunning tanks operate at pH levels of about 5.5–6.0, corresponding to CO_2_ levels of 200–450 mg L^−1^ [19,20]. Subsequently, water was flushed with oxygen to avoid anoxic water. When an oxygen concentration of 8.5 mg/L was obtained, halibut were immersed in the water for stunning. After 446 ± 32 s of immersion in the CO_2_, twenty-eight fish were bled by applying a double gill-cut after carbon dioxide stunning. One fish was not bled to observe the consistency of unconsciousness.

One fish was euthanized for head dissection to determine the exact position of the brain. The needle-shaped electrodes (2 cm length, 1.5 cm diameter, 55% silver, 21% copper and 24% zinc) were placed percutaneously across the skull and remained on the surface of the brain lateral to *opticum tetum*. For handling, prior to electrode insertion, fish were restrained on a rectangular wooden plate (25 × 40 cm) using adjustable straps around the body. The first electrode was inserted at 4 cm caudal from an imaginary line between the eyes. The next electrode was inserted at 0.5 cm ventral to the first electrode. The third intramuscular earth electrode was inserted in the tail. Cyanoacrylate under the skin was used to fix the electrodes. This methodology is similar to that reported by [23].

### 2.2. EEG Analyses

EEG was recorded using a Bioamplifier connected to a PowerLab 8/35 from AD instruments, with 1000 Hz sampling frequency. Baseline EEG was recorded during 60 s before stunning in a tank with fresh seawater (50 × 35 × 23 cm) (*n* = 29 fish). After the baseline recording, fish were placed in the stunning tank with CO_2_-saturated seawater, which was aerated prior to stunning of halibut. In addition to the EEG recording, behaviour was recorded using a camcorder. Behaviour of the restrained fish was used to identify escape attempts and movement artefacts to support interpretation of the EEG traces. Escape attempts were characterized by strong muscle contractions in an attempt to escape after immersion in the CO_2_ tank. The proportion of fish with escape attempts, latency (s) for onset movements from the immersion in the CO_2_ tank and duration of movements were recorded.

EEG recordings were monitored, saved and analysed using LabChart 7 Pro (version 7.3.7, AD Instruments, Cologne, Germany). Periods with clear artefacts lasting for more than a few seconds were identified and discarded in the subsequent analysis. Thereafter, on LabChart, the EEG recordings were filtered (band pass: 0.1 to 45 Hz) for visual analysis such as filtering delate background noise. The screen was split in two in order to compare each EEG portion from the baseline of each fish (Figure 1, Figure 2, Figure 3, Figure 4 and Figure 5). Visual inspection of the traces was used to assign portions of the EEG to one of the 4 phases with the following characteristics:

‘Normal’: activity identical to the baseline but after immersion in the CO_2_ tank;

‘Transitional’: increase in the amplitude and similar frequency to baseline phase;

‘Suppressed’: a greatly suppressed EEG but containing low amplitude and low frequency wave activity; such a phase can be identified by the reduction of more than 40% on the amplitude in comparison with the baseline;

‘Iso-electric’: residual low-level noise indicating lack of EEG activity.

Normal, transitional, suppressed and iso-electric patters have been reported previously in studies with chickens stunned with CO_2_ [16,24].

Representative 2 s epochs free of noise and/or artefacts were selected from EEG trace for spectral analysis. Six fish (6/29) were discarded due to artefacts on the signal, preventing the selection of several 2 s epochs on the baseline. Hence, spectral analysis was performed on 23 fish registers. Spectral analyses were performed on filtered recordings (band pass: 4 to 45 Hz) was set: FFT 1K; data window Hamming and window overlap 93.75%. Such filter diluted the interference from breathing. The percentage of total power was determined from 2 s epochs for the following wave bands of contribution to total power: Delta (0–4 Hz), Theta (4–8 Hz), Alpha (8–12 Hz) and Beta (12–32 Hz). In addition, Total power (v^2^) (Ptot), median frequency (Hz) (F50) and spectral edge frequency 95% (Hz) (SEF) were calculated from each 2 s epoch and classified according to the EEG phases identified on visual analysis. Epochs of less than 2 s were not included.

Five criteria were considered for fish to define unconscious: decrease of Ptot, decrease of F50, decrease of SEF, increase of low frequencies (Delta and Thera) (0–8 Hz) and decrease of high frequencies (Alpha and Beta) (8–32 Hz). Such criteria have been used in birds [25], lambs [26], cattle [27,28] and fish [13,29]. However, these studies did not evaluate all the above-cited criteria used in the present study. Changes of the EEG signal to a transitional, suppressed or iso-electric phase were based on visual comparison of the signal with the baseline.

### 2.3. Statistical Analyses

The variables of escape attempts (proportion of fish with escape behaviour), latency (s) for onset movements from immersion in CO_2_ tank and duration (s) of movements are shown with descriptive statistics. The effect of time process on Ptot, F50, SEF, low frequencies (LF) and high frequencies (HF) were analysed through linear mixed models. In this case, visual phase was included as a fixed factor with five levels (baseline, transitional, excitation, suppressed and iso-electric), and a normal distributed random effect of fish was included to adjust the variance between different fish and correlation between measurements from a single fish. Results for each fish in each time were first summarized by the sample median, and it was considered as the response variable for the linear mixed models. The adequacy of fitted models was assessed through residual analysis. Ptot values were log-transformed due to high skewness, and back-transformed when the model results were presented. The results provided by the fitted model were summarized through estimated means and corresponding 95% confidence intervals. Furthermore, the variance of random effects and corresponding intra-class correlation coefficient (ICC) are presented as well. Statistical significance (*p* < 0.05) was verified for all response variables, and the analysis was succeeded by a Tukey multiple comparison post hoc test, where the conclusions are based on a 5% global significance level. All analyses were performed using the R environment for statistical computation, version 4.1.2 [30]. The lme4 library [31] was used in fitting the regression models.

## 3. Results

During immersion in CO_2_-saturated seawater, a series of consistent changes in the appearance of the EEG were observed: baseline (Figure 1), a transitional phase (Figure 2), a phase with high-amplitude–high-frequency signal (excitation phase) (Figure 3), a suppressed phase (Figure 4) and finally an iso-electric phase (Figure 5). Escape attempts were noted in 10 out of 29 (34.5%) fish with a median latency of 12.5 ± 17.5 s from immersion in CO_2_ tank and duration of 14 ± 10.5 s (median ± Interquartile Range IQR) (Figure 6).

Timings for onset and duration of the different phases observed in EEG recordings are presented in Table 1. Both suppressed and iso-electric phases were not observed in all fish after bleeding. Furthermore, one fish did not show suppressed phase before bleeding; for this reason, the number of fish was not the same in all phases. The characteristics of EEG from each fish can be observed in Figure 7. The transitional phase was the first change observed on EEG and was followed by an excitation phase. It is possible that consciousness was not lost at the start of the excitation phase. The mean time for the onset of the suppressed phase was 258.8 ± 46.2 s. The visual characteristics of the suppressed phase suggest that halibut is unconscious at this point.

The results from the model used to understand the spectral analysis can be observed in Table 2. The values on the excitation phase were higher in Ptot, F50 and HF due to both high amplitude and high frequency on the trace (See Figure 3). Compared to the baseline values, Ptot value was lower at the iso-electrical phase but not at the suppressed phase. However, the values of F50, SEF and HF in the suppressed phase were significantly lower in contrast to baseline, in concordance with an unconscious animal. The LF values were significantly higher at both suppressed and iso-electric phases, compared to baseline. F50, SEF, LF and HF values were similar between the suppressed and iso-electric phase.

## 4. Discussion

The results from spectral analysis, such as the reductions in F50, SEF and HF and the increase in LF in the suppressed phase, are compatible with the unconsciousness state, with both LOC and loss of pain sensibility. The variables of spectral and visual analysis can be used together to determine the onset of unconsciousness in fish. The main challenge in recording baselines in conscious fish was a predominance of high frequencies, which compromises the spectral analysis. Refining the handling protocol, using local anaesthesia before implanting electrodes, and reducing stress during handling can improve the quality of baseline records.

After the excitation phase, fish showed a suppression phase (Figure 4). The suppressed phase can be explained by marked depression caused by high concentrations of CO_2_ in the cerebrospinal fluid [32]. We interpret the suppressed phase as typical for loss of consciousness, as the spectral analysis showed a significant decrease in F50, SEF and HF, in comparison to baseline values compatible with an unconscious animal as reported in broilers. Broilers submitted to general anaesthesia presented a reduction in F50 and SEF in comparison to baseline values [17]. In hens killed using CO_2_, the suppressed EEG which followed the transitional phase had a very distinctive appearance and spectral properties were strongly associated with an unconscious state [24]. Fish stunned by an electrical current or by percussion showed a period of strong depression of the electrical activity on the EEG, after the epileptiform insult labelled in this study as “exhaustion phase” [33]. Hence, the subsequent phases observed in the EEG analysis of halibut in our study exposed to saturated CO_2_ showed a pattern consistent with a progressive induction of unconsciousness, and the LOC can be estimated on the onset of suppressed phase due to changes on spectral variables.

Bowman et al. [34] found that for rainbow trout (*Oncorhynchus mykiss*) immersed in CO_2_-saturated water at different acclimation temperatures of 14, 8 and 2 °C, it took significantly longer to lose equilibrium after submersion in CO_2_-saturated water at 2 °C compared to 14 °C (*p* < 0.05). Furthermore, the eye-roll reflex was lost shortly after equilibrium, taking (mean, min-max value) 1.30 (1–2) min, 3.15 (2–4.5) min and 4.15 (3–5.5) min at 14, 8 and 2 °C, respectively, and the time taken for the eye-roll reflex to be lost was significantly different between all temperature groups [34]. Breathing ceased at 3.61 (2.90–4.55) min at 14 °C, 5.74 (4.01–7.33) min at 8 °C and 9.00 (7.33–11.33) min at 2 °C, and the time taken for breathing to cease was significantly different between all temperature groups [34].

In pigs exposed to 80% or 95% CO_2,_ time to loss of consciousness was 46 s and 33 s, respectively [35]. In poultry, McKeegan et al. [36] reported a suppressed pattern in broiler chickens exposed to 40% CO_2_ + 60%. Nitrogen was observed to occur at 23 ± 4 s. Suppressed EEG occurred on average at 60 ± 23 s in broiler chickens exposed to CO_2_ in a concentration below 40% in a multistage system [37]. Such times indicate a faster induction process in birds and pigs than fish, when CO_2_ or CO_2_ mixtures are used. Concentrations of 80% and 90% CO_2_ in air for stunning are equivalent to 800,000 parts per million (ppm) and 900,000 ppm, respectively [38]. In the atmosphere, CO_2_ is normally found at concentration between 300 and 700 ppm. In seawater, CO_2_ concentrations of 200 to 400 mg L^−1^ for stunning are equivalent to 200 to 400 ppm, respectively. Additionally, pigs and birds are exposed to high CO_2_ concentrations and hypoxic environments (<12% O_2_ in air), different from our experiment in which the O_2_ concentrations in water were at normal farm system levels. Therefore, fish stunning with CO_2_ using both lower CO_2_ concentrations in water and a normoxic environment is contrary to CO_2_ stunning in pigs or birds. This might explain the reason why the onset of unconsciousness is faster in pigs and birds than in fish.

Another possible explanation for the longer times to loss of consciousness compared to broilers is related to the Bohr effect. The Bohr effect generally describes the effect of pH on the blood–O_2_-binding affinity. In most teleost fish, the Bohr effect is twice as strong as in mammals because up to four additional protons are bound, thereby stabilizing the low-O_2_-affinity T-state [39]. T-state haemoglobin (Tense haemoglobin) form is the most stable haemoglobin form in the absence of oxygen [40]. The presence of a Bohr effect allows more-efficient blood–O_2_ transport because under the acidic conditions in the tissue capillaries, where CO_2_ or lactic acid are released into the blood, the Hb–O_2_ affinity is decreased. The right-shift of the O_2_-binding curves allows increased offloading of O_2_ without compromising venous *P*O_2_; thereby the O_2_-diffusion gradient from the blood to the tissues can be maintained [39]. The affinity of haemoglobin for O_2_ is quantified with P_50_, which is the *P*O_2_ at 50% saturation. In most birds, and in contrast to mammals, there is no independent effect of CO_2_ on P_50_. CO_2_ forms carbamino compounds with haemoglobin in mammals, and these cause small increases in P_50_. In some birds, such as sparrows and burrowing owls, the Bohr effect is greater when pH is changed with CO_2_ compared with fixed acid. Therefore, carbamino formation does occur and can decrease O_2_ affinity in stripped avian haemoglobin [41]. Therefore, as this mechanism means that fish are more resistant to environments with high concentrations of CO_2_, perhaps the maintenance of the pH of the cerebrospinal fluid may be more efficient in fish, explaining the slower induction of unconsciousness in comparison with birds or mammals.

The iso-electric phase was observed after bleeding (Figure 5). This phase corresponds to low-level noise reflecting a lack of EEG activity as was noted in broilers under low-atmospheric-pressure stunning [25] and during exposure to gas mixtures [42]. The iso-electric phase, in conjunction with other indicators such as the absence of reflexes and cardiac arrest, are indicators of brain death in birds under anaesthesia [17]. High concentrations of CO_2_ in the brain, in addition to hypovolemia and ischemia caused by cutting the gills, suggest that the iso-electric phase is an indicator of brain death.

Given that the visual analysis from the EEG is a qualitative and subjective analysis, we performed the spectral analysis on the EEG data in Halibut. In other species, the spectral analysis from EEG traces was performed with animals under anaesthesia [17,43,44]. This approach allows the administration of various drugs that enhance analgesia, anaesthesia, muscle relaxation, hypnosis and amnesia responses, thus obtaining EEG recordings free from any interference or artefacts.

Regarding Ptot, we did not observe significant differences among the baseline and suppressed phases; the difference was observed at the iso-electrical phase. In rainbow trout (*Oncorhynchus mykiss*) submerged in CO_2_ at pH < 5.0 and 10 °C, EEG signal amplitude (Ptot) increased immediately after transfer to the treatment tank compared to pre-treatment amplitude measurements. Time until amplitude declined to <50% pre-treatment amplitude was 7.44 (4–15) min after transfer [35].

Sandercock et al. [17] reported an increase in Ptot (uV^2^) during sedation and deep anaesthesia, using dexmedetomidine 80 µg/kg IM and sevoflurane 8% vaporizer, respectively. Anaesthesia using propofol in turkeys induced a reduction in Ptot from EEG [44]. The progressive increase in Ptot and subsequent fall was also reported in broilers during exposure to low-atmospheric-pressure stunning [25] and hens under anaesthesia. In rainbow trout (*Oncorhynchus mykiss*) subjected to anaesthesia with MS-222 [29], a decrease in Ptot was observed from 4 min of immersion in the anaesthetic tank.

Values to F50 and SEF were significantly lower at the suppressed phase in comparison with values from baseline, a similar result reported by Hernandez et al. [45] and Sandercock et al. [17] for poultry under anaesthesia. Bowman et al. [29] did not find significant differences on F50 over the time of anaesthesia of rainbow trout when compared to baseline. Bowman et al. [34] did not observe a significant difference between pre-treatment mean and treatment values from EEG regarding F50 in rainbow trout (*Oncorhynchus mykiss*) submerged in CO_2_ water at pH < 5.0 and 10 °C over 40 min. Hence, F50 and SEF can be indicators in future studies with spectral analysis from EEG in fish. The reduction in F50 and SEF is comparable to the shift from the conscious to unconscious state, when the high frequencies (8–32 Hz) are reduced and the low frequencies (0–8 Hz) are predominant. F50 and SEF seem to be more robust for identifying unconsciousness in fish. Hence, the characteristics of the visual inspection (subjective analysis) are consistent with the observed differences in the variables F50 and SEF from spectral analysis (objective analysis) in halibut exposed to CO_2_-saturated seawater.

Low frequencies (<8 Hz) waves were specifically examined, as they are associated with sleep, anaesthesia and LOC. Such increases of low frequency activity was also reported in broilers under low-atmospheric-pressure stunning [25]. In rainbow trout (*Oncorhynchus mykiss*) submerged in CO_2_ water at pH < 5.0 and 10 °C over 40 min, Bowman et al. [34] showed no significant difference between the pre-treatment means and the relative power of high frequencies (Beta/Alpha) and low frequencies (Theta/Delta) over treatment. Bowman et al. [29]) did not find significant differences between low or high frequencies over the time of anaesthesia of rainbow trout when compared to baseline. In eels, the use of a captive needle pistol immediately induced low-frequency waves tending to no brain activity recorded on EEG [46,47]. Hence, low-frequency activity on suppressed and iso-electric phases suggests a deep state of unconsciousness in halibut from immersion in CO_2_-saturated seawater. Results of frequency bands as determined by spectral analysis suggest that halibut immersed in CO_2_-saturated seawater lose consciousness at 258.8 ± 46.2 s from the immersion. In our results, halibut showed an increase in low frequencies during the suppressed phase in comparison with the baseline, indicating a state of unconsciousness at the suppressed phase.

Further, high-frequency (8–32 Hz) activities were reduced in the suppressed phase in halibut in contrast with the baseline. Results from spectral analysis of F50, SEF, low-frequency and high-frequency values on the suppressed phase are in concordance with the shift of frequencies from an unconscious animal. Hence, in halibut, the onset for unconsciousness was at 258.8 ± 46.2 s from immersion in CO_2_-saturated seawater.

Regarding the escape attempts, fish were restrained during the stunning procedure; such conditions are not suitable to assess behaviour, but strong muscle contractions can still be observed. The number of escape attempts would be probably higher in unrestrained fish. In terrestrial animals, inhalation of concentrations above 30% CO_2_ by volume in atmospheric air causes aversion, irritation of membranes, pain and respiratory distress during the induction phase in pigs and poultry [48]. Trout, salmon, carp and eel exposed to CO_2_ displayed vigorous aversive reactions, rapid swimming and attempts to escape [22]. Rainbow trout (*Oncorhynchus mykiss*) submerged for 12 min in CO_2_-saturated water at pH < 5.0, and at different acclimation temperatures of 14, 8 and 2 °C, all showed strong aversive behaviours in the form of repeated attempts to escape the tank and vigorous swimming. The observations made in halibut in the current study had a similar pattern of reactions to high CO_2_ concentrations.

Transitional, suppressed and iso-electric phases have been reported in hens, broilers, ducks and turkeys exposed to N_2_- or CO_2_-filled foam [16] and hens killed by CO_2_ [24]. Such different phases were also described in experiments with pigs exposed to high CO_2_ levels [35]. In halibut, shortly after immersion in the stunning tank, movements and muscular contractions related to the escape attempts and breathing movements from gills cause artefacts on the recorded signals and therefore influence the assessment of the EEG signal especially on the transitional phase. Similar to our results, the time for the onset of the transitional phase in hens exposed to N_2_ foam was 10 s [16].

However, the duration of the transitional phase in hens was only 20 s, which is considerably shorter than in our results (108.8 ± 54.9 s). The transitional phase (Figure 2) is probably a transition trace between the baseline and the excitation phase. The EEG visual analysis alone is not sufficient to determine the onset of unconsciousness. Therefore, spectral analysis helped in confirming the onset of unconsciousness at the beginning of the suppressed phase.

After the transitional phase, a period of consistent high amplitude and high frequency (Figure 3) was observed. This is called the excitation phase, since the characteristics observed for this phase were very similar to those of a general epileptiform or grand mal insult in studies on fish stunned by electric shock or percussion [33,49]. It is possible that consciousness was not lost at the start of the excitation phase. Hence, results from the spectral analysis suggest that halibut lost consciousness at the beginning of the suppressed phase, 258.8 ± 46.2 s, due to reduction in F50, SEF and % contribution from HF. Epileptic-like EEG patterns or general epileptiform insults or excitation phase were not reported in hens, broilers, ducks and turkeys exposed to N_2_- or CO_2_-filled foam [16], hens killed by CO_2_ [24] or broilers killed by gas mixtures [48]. The gas stunning process induces unconsciousness faster in pigs and birds, <50 s; thus, the excitation phase or general epileptiform insult were probably not observed in these species. The excitation phase could be masked by movement artefact, for example.

Similar to our results, after the transitional phase, Mckeegan et al. [16] reported a suppressed phase at 30 s after the hens were immersed in the N_2_ foam, which is a reliable indicator of LOC. Broilers exposed to 2% residual oxygen in a mixture of 20% carbon dioxide and 80% argon in air showed rapid loss of spontaneous electrical activity in the first 10 s of exposure, leading to a profoundly suppressed EEG (LOC) at the onset of loss of posture at 12 s. The magnitude of EEG suppression occurring prior to the onset of convulsions at 17 s is indicative of a pathological brain state that is incompatible with consciousness and sensibility [50].

Previous studies on EEG in fish immersed in CO_2_-saturated water are scarce [11,34], and the available studies used the time to loss of visual evoked responses as a measure of (un)consciousness rather than EEG recordings. Bowman et al. [34] studied rainbow trout (*Oncorhynchus mykiss*) submerged in CO_2_ water at pH < 5.0 and 10 °C over 40 min, showing that there is a poor relationship between the loss of visual evoked responses and loss of visual indicators of consciousness, which suggests that when visual indicators alone are used, fish risk being misjudged as insensible before sensibility is lost. To our knowledge, there is no manuscript measuring EEG activity in halibut immersed in CO_2_-saturated water and comparing the visual and the spectral variables. Our results can help to produce specifical requirements to stun/kill farmed fish according to the Annex 1 of Regulation 1099/2099 in Europe.

## 5. Conclusions

Stunning of halibut in CO_2_-saturated seawater, which is well-oxygenated, induces vigorous movements and escape attempts. The progressive induction of unconsciousness showed four different phases on EEG. Spectral analysis from EEG traces allowed confirming the phases observed by visual inspection, namely the transitional, excitation, suppressed and iso-electric phases. The onset of unconscious was determined at the suppressed phase. Visual EEG analyses in conjunction with spectral analysis allowed a more accurate approach to assess unconsciousness.

In order to decrease suffering at the time of slaughter, the induction of unconsciousness must be immediate (<1 s) or progressive, but not aversive. In our experiment, halibut took 258.8 ± 46.2 s to lose consciousness. As movements and clear escape attempts, indicative as being aversive, were also observed, the use of CO_2_ in seawater is not recommended for stunning of Atlantic halibut.

## Figures and Tables

**Figure 1 animals-13-01993-f001:**
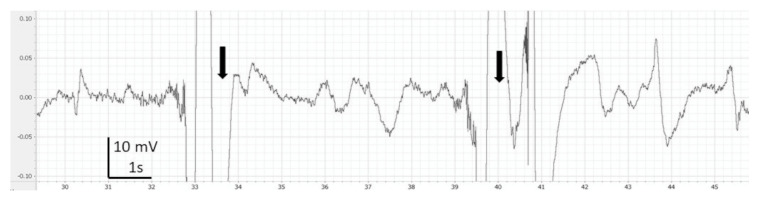
Example of baseline (normal EEG) observed in halibut before immersion in CO_2_-saturated seawater. Band pass filter (0.1–45 Hz). Arrows represent breathing interference.

**Figure 2 animals-13-01993-f002:**
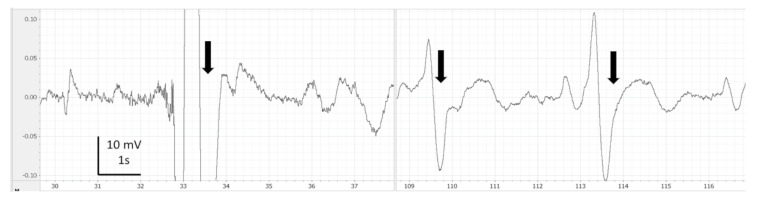
Example of comparison of baseline (left side) vs. transitional phase (right side) observed in halibut from immersion in CO_2_-saturated seawater. Band pass filter (0.1–45 Hz). Arrows represent breathing interference.

**Figure 3 animals-13-01993-f003:**
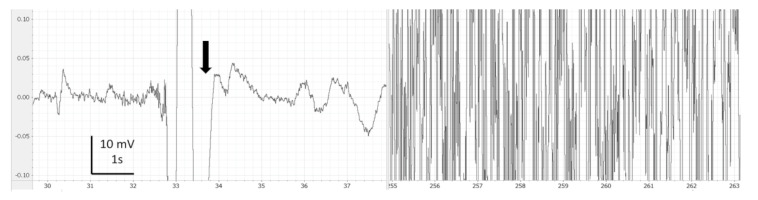
Example of comparison of baseline (left side) vs. excitation phase (right side) observed in halibut from immersion in CO_2_-saturated seawater. Band pass filter (0.1–45 Hz). Arrows means breathing interference.

**Figure 4 animals-13-01993-f004:**
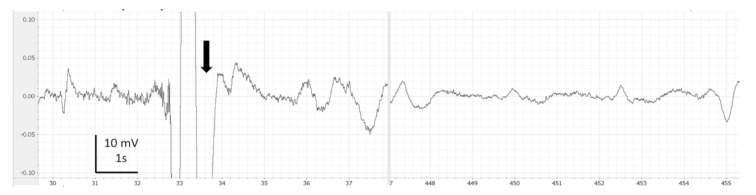
Example of comparison of baseline (left side) vs. suppressed phase (right side) observed in halibut from immersion in CO_2_-saturated seawater. Band pass filter (0.1–45 Hz). Arrows represent breathing interference.

**Figure 5 animals-13-01993-f005:**
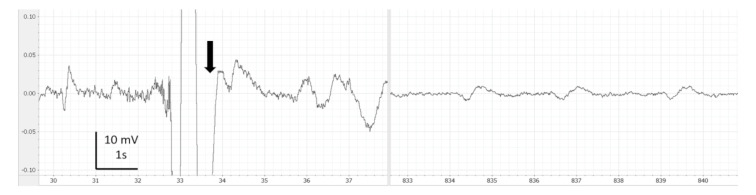
Example of comparison of baseline (left side) vs. iso-electric phase (right side) observed in halibut from immersion in CO_2_ saturated seawater. Top of the figure with band pass filter (0.1–45 Hz).

**Figure 6 animals-13-01993-f006:**
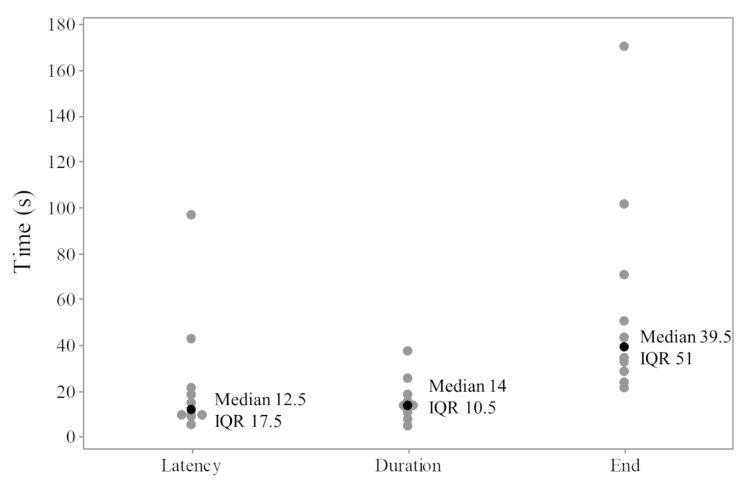
Individual time (s) for escape attempts from immersion in CO_2_-saturated seawater. Latency for first movement, duration of movements and end of last movement. The green point indicates individual values; the black points represent the median value. Interquartile range IQR.

**Figure 7 animals-13-01993-f007:**
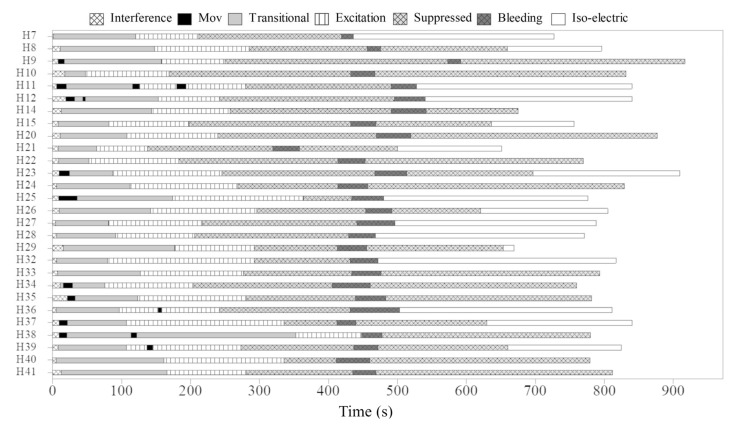
Characteristics of the EEG trace in halibut (*n* = 28) immersed in CO_2_-saturated seawater.

**Table 1 animals-13-01993-t001:** Onset and duration (s) (mean ± SD) of different EEG phases observed in 29 halibuts from immersion in CO_2_-saturated seawater.

	Transitional	Excitation Phase	Suppressed	Suppressed *	Iso-Electric *
Onset	13.7 ± 10.3	126.2 ± 55.0	258.8 ± 46.2	482.6 ± 32.4	567.4 ± 89.1
Duration	107.9 ± 54.3	134.8 ± 36.5	180.2 ± 65.6	251.4 ± 83.7	237.5 ± 94.1
*n*	29	29	28	19	17

* After bleeding (time after CO_2_ immersion).

**Table 2 animals-13-01993-t002:** Summary of the fitted linear mixed models for the spectral analysis (Total Power: Ptot; Median Frequency: F50; Spectral Edge Frequency: SEF, contribution from Low Frequencies (0–8 Hz): LW; contribution from High Frequencies (8–32 Hz): HF, presented as estimated means (95% confidence interval) in halibut from immersion in CO_2_-saturated seawater. Different letters at the same column indicate different means according to the Tukey multiple comparison post hoc test.

Phase	Ptot (µV^2^)	F50 (Hz)	SEF (Hz)	LF (%)	HF (%)
Baseline	21.4 (11.7; 39.0) ^B^	10.5 (9.2; 11.7) ^B^	31.6 (29.5; 33.7) ^A^	38.9 (31.9; 45.8) ^B^	55.4 (49.3; 61.6) ^B^
Transitional	32.98 (18.08; 60.2) ^B^	10.0 (8.7; 11.2) ^BC^	32.3 (30.2; 34.4) ^A^	41.9 (34.9; 48.8) ^B^	51.9 (45.7; 58.1) ^BC^
Excitation	5729.9 (3141.22; 10,452.1) ^A^	12.1 (10.9; 13.4) ^A^	32.7 (30.6; 34.8) ^A^	25.5 (18.5; 32.5) ^C^	68.4 (62.2; 74.5) ^A^
Suppressed	28.42 (15.58; 51.8) ^B^	8.1 (6.8; 9.3) ^D^	24.2 (22.0; 26.3) ^B^	52.7 (45.7; 59.7) ^A^	45.0 (38.8; 51.2) ^C^
Iso-electric	5.11 (2.39; 10.9) ^C^	8.3 (6.9; 9.8) ^CD^	26.3 (23.7; 28.9) ^B^	53.2 (44.9; 61.5) ^A^	42.7 (35.4; 50.0) ^C^
Random effect	21.4 (11.7; 39.0) ^B^	10.5 (9.2; 11.7) ^B^	31.6 (29.5; 33.7) ^A^	38.9 (31.9; 45.8) ^B^	55.4 (49.3; 61.6) ^B^

## Data Availability

Data are contained within the article.

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
