# Peer review of "Pre-Slaughter Stunning of Farmed Atlantic Halibut in CO2-Saturated Seawater: Assessment of Unconsciousness by Electroencephalography (EEG)"

_animals, 2023, doi:10.3390/ani13121993_

Round 1

Reviewer 1 Report

Congratulations to the authors for addressing a very important issue in a rigorous manner. Below I provide some comments that I hope are of help in order to further improve the paper.

Ll. 46-51 – This detailed description is not relevant to the main topic of the paper. If the authors want to point out that it is likely that the aquaculture production of the species will grow in the coming years, I would suggest they simply mention that “Atlantic halibut is a species of high value in the market that is likely to play a more important role in European aquaculture production“, or something along these lines. 

Ll. 64-65 –  I could explicitly mention that the animal must be killed before regaining consciousness: “The second step is killing by bleeding out or the application of another method to induce death while the fish is still unconscious”. 

L. 308 – Perhaps “Another possible explanation for the longer times to loss of consciousness compared to broilers” is clearer. 

Ll. 310-313 – It would be great if the authors could also briefly talk about the characteristics of the Bohr effect in birds.

Ll. 329-330 – Perhaps “Given that the visual analysis from the EEG is a quantitative and subjective analysis, we performed the spectral analysis on the EEG data in Halibut” is clearer to the reader.

Ll. 49-451 – Fish welfare can be good or bad, therefore “for meet welfare” is not grammatically wrong but also a confusing concept in this sentence – where the authors explain that the goal is to provide the animals with good welfare at slaughter. A sentence like “In order to decrease suffering at the time of slaughter, the induction of unconsciousness must be immediate (< 1s) or progressive but not aversive” probably works better. I would not suggest using the concept of good welfare here, since there are other aspects of the slaughter process (e.g., handling) that are likely to prevent these animals from experiencing good welfare at this stage of the production cycle.

Author Response

We are pleased to submit the revised version of animals-2382721 Pre-slaughter stunning of farmed Atlantic halibut in CO2 saturated seawater: assessment of unconsciousness by electroencephalography (EEG). We appreciate the constructive suggestions of the reviewers and have addressed each of their comments as outlined below.

Best regards, 

The authors,

Ll. 46-51 – This detailed description is not relevant to the main topic of the paper. If the authors want to point out that it is likely that the aquaculture production of the species will grow in the coming years, I would suggest they simply mention that “Atlantic halibut is a species of high value in the market that is likely to play a more important role in European aquaculture production“, or something along these lines. 

Authors: The suggestion made by the reviewer was incorporated in the text on line 46.

Ll. 64-65 –  I could explicitly mention that the animal must be killed before regaining consciousness: “The second step is killing by bleeding out or the application of another method to induce death while the fish is still unconscious”.

Authors: The suggestion made by the reviewer was incorporated in the text on line 61.

  1. 308 – Perhaps “Another possible explanation for the longer times to loss of consciousness compared to broilers” is clearer.

Authors: The suggestion made by the reviewer was incorporated in the text on line XX

Ll. 310-313 – It would be great if the authors could also briefly talk about the characteristics of the Bohr effect in birds.

Authors: The suggestion made by the reviewer was incorporated in the text on line 317-322. The affinity of hemoglobin for O2 is quantified with P50, which is the PO2 at 50% saturation. In most birds, and in contrast to mammals, there is no independent effect of CO2 on P50. CO2 forms carbamino compounds with hemoglobin in mammals, and these cause small increases in P50. In some birds, such as sparrows and burrowing owls, the Bohr effect is greater when pH is changed with CO2 compared with fixed acid. Therefore, carbamino formation does occur and can decrease O2 affinity in stripped avian hemoglobin.

Ll. 329-330 – Perhaps “Given that the visual analysis from the EEG is a quantitative and subjective analysis, we performed the spectral analysis on the EEG data in Halibut” is clearer to the reader.

Authors: The suggestion made by the reviewer was incorporated in the text on line 334. Given that the visual analysis from the EEG is a qualitative and subjective analysis, we performed the spectral analysis on the EEG data in Halibut

 Ll. 49-451 – Fish welfare can be good or bad, therefore “for meet welfare” is not grammatically wrong but also a confusing concept in this sentence – where the authors explain that the goal is to provide the animals with good welfare at slaughter. A sentence like “In order to decrease suffering at the time of slaughter, the induction of unconsciousness must be immediate (< 1s) or progressive but not aversive” probably works better. I would not suggest using the concept of good welfare here, since there are other aspects of the slaughter process (e.g., handling) that are likely to prevent these animals from experiencing good welfare at this stage of the production cycle.

Authors: The suggestion made by the reviewer was incorporated in the text on line 454. In order to decrease suffering at the time of slaughter, the induction of unconsciousness must be immediate (< 1s) or progressive but not aversive.

Thank you, 

Sincerely

Reviewer 2 Report

The work addresses a very relevant aspect in the observance and guarantee of animal welfare during the stunning and slaughter processes. The justification is pertinent, the methodology adequate and the results consistent. All this supports incontestable conclusions.

I can only congratulate the authors and urge the editor for its publication.

Author Response

Dear reviewer, 

Thank you for your time to read our manuscript. 

Sincerely, 

The authors, 

Reviewer 3 Report

The use of stunning methods previous the slaughter is a major scientific issue that concerns scientists. However, despite this study being well presented, the specific issue does not meet the criteria of innovative research. 

------------------

There is no dough that humane treatment during slaughter is a very important scientific issue that remains unclear. To protect animals at slaughter, they should be rendered unconscious and insensible by stunning to avoid pain, fear, or distress before slaughter, which is a general provision in the EU legislation to protect animals at slaughter.  This statement is confirmed and highlighted in the present study. However, the manuscript doesn’t support this direction. The study concludes that CO2-saturated water is not recommended for pre-slaughter handling of halibut due to the extended time required for the onset of unconsciousness and the observed aversive behavior. 

In general, the study is well-designed and well-presented. Comments concerning the details of the research are not important, as the reason for rejection is a strong disagreement concerning the null hypothesis which is in dispute with the scientific input. The present study has confirmed that the use of carbon dioxide is a non-acceptable stunning method. The savageness of this technique is the reason why it doesn’t meet the welfare criteria of farewell and is not accepted according to EU legislation. So the conclusion of the present study was already known and was expected. It would be really a challenge and innovation if the results of the present study were contrary to what was expected and tended towards fish welfare. However, as it is presented, it doesn’t meet the criteria of innovative research that will offer useful scientific information.  

The manuscript is well written in the English language. 

Author Response

We are pleased to submit the revised version of animals-2382721 Pre-slaughter stunning of farmed Atlantic halibut in CO2 saturated seawater: assessment of unconsciousness by electroencephalography (EEG). We appreciate the constructive suggestions of the reviewers and have addressed each of their comments as outlined below.

There is no dough that humane treatment during slaughter is a very important scientific issue that remains unclear. To protect animals at slaughter, they should be rendered unconscious and insensible by stunning to avoid pain, fear, or distress before slaughter, which is a general provision in the EU legislation to protect animals at slaughter. This statement is confirmed and highlighted in the present study. However, the manuscript doesn’t support this direction. The study concludes that CO2-saturated water is not recommended for pre-slaughter handling of halibut due to the extended time required for the onset of unconsciousness and the observed aversive behavior.

In general, the study is well-designed and well-presented. Comments concerning the details of the research are not important, as the reason for rejection is a strong disagreement concerning the null hypothesis which is in dispute with the scientific input. The present study has confirmed that the use of carbon dioxide is a non-acceptable stunning method. The savageness of this technique is the reason why it doesn’t meet the welfare criteria of farewell and is not accepted according to EU legislation. So the conclusion of the present study was already known and was expected. It would be really a challenge and innovation if the results of the present study were contrary to what was expected and tended towards fish welfare. However, as it is presented, it doesn’t meet the criteria of innovative research that will offer useful scientific information. 

Authors: We appreciate the Reviewer’s comments. 

The use of CO2 in water to stun/kill fish is not recommended by the World Organisation for Animal Health – OIE. However, the use of CO2 is still common to kill halibut in Norway. Additionally, on the EU Regulation on the protection of animals at the time of killing (1099/2009), Chapter I, Article 1 Subject matter and scope it is stated that “This Regulation lays down rules for the killing of animals bred or kept for the production of food, wool, skin, fur or other products as well as the killing of animals for the purpose of depopulation and for related operations”. Regarding to fish, “only the requirements laid down in Article 3(1) shall apply”. Chapter II, Article 1 points that ” Animals shall be spared any avoidable pain, distress or suffering during their killing and related operations”. However, the same Regulation indicates that “Recommendations on farm fish are not included in this Regulation because there is a need for further scientific opinion and economic evaluation in this field”. Therefore, there is gap of real protection of fish or a list of stunning methods at the time of killing (Annex 1). From the authors´ experience, very few fish species in selected countries are stunned before bleeding. The last publications from the European Food Safety Authority (EFSA) show critical points for poor animal welfare before slaughter in poultry stunned by waterbath stunning systems (EFSA, 2019).

  • Hanging upside down is a physiologically abnormal posture for poultry; inversion, and shackling are practices that cause pain and fear in conscious birds (ToR-2).
  • In electrical waterbath stunning, not all birds processed at the same time receive the same current. Therefore, due to the electrical settings usually used, some birds don’t receive sufficient current to become unconscious (ToR-2). https://www.efsa.europa.eu/en/efsajournal/pub/5849

Concerning pigs, the conclusion is the same; the actual method of CO2 at high concentrations and electrical stunning used at slaughterhouses do not meet the criteria of humane slaughter (EFSA, 2020).

  • Electrical and mechanical (excluding firearms) stunning methods require restraint that may impose per se additional pain and fear. Such welfare consequences will persist during the straining period until successful stunning.
  • Movement of pigs into a single line for the purpose of electrical stunning will cause fear and pain worsened by the use of force (e.g. electric goads). The welfare consequences will be exacerbated with increased throughput rates.
  • Exposure to CO2 at high concentrations is considered a serious welfare concern by the Panel, because it is highly aversive and causes pain, fear and respiratory distress. https://doi.org/10.2903/j.efsa.2020.6148

According to the literature, stunning methods should induce immediate loss of consciousness. If loss of consciousness is not immediate, the induction of unconsciousness should not cause avoidable pain and suffering in animals.

Hence, even in terrestrial species that have been extensively studied the current criteria for humane slaughter are not met in commercial slaughterhouses, so that in farmed fish we could expect a worse scenario in terms of negative animal welfare consequences at slaughter.

Our study brings interesting results in a very neglected species, using new approaches to analyse EEG records. This study can be used to regulate pre- and slaughter procedures in farmed fish in the future.

Sincerely, 

The authors,

Round 2

Reviewer 3 Report

The authors' points are entirely understandable. The gap in the legislation and the fish protection towards welfare is a fact. However, the present study doesn’t give information supporting that CO2-saturated water could be recommended for pre-slaughter handling of halibut to avoid pain, distress, or suffering during their killing and related operations. Therefore the scientific information presented are not applicable and usable. 

Good quality English text.